# EFFICIENT BLOCKWISE DIVERSE ACTIVE LEARNING

## ABSTRACT

Active learning (AL) techniques are known for selecting the most informative data points from large datasets, thereby enhancing model performance with fewer labeled samples. This makes AL particularly useful in tasks where labeling is limited or resource-intensive. However, most existing effective methods rely on uncertainty scores to select samples, often overlooking diversity, which results in redundant selections, especially when the batch size is small compared to the overall dataset. This paper introduces Efficient Blockwise Diverse Active Learning (EBDAL), a generalizable framework that combines uncertainty with diversity-based selection to overcome these limitations. By partitioning the dataset into blocks via a clustering strategy, we ensure diverse sampling within each block, enabling more efficient handling of large-scale datasets. To quantify diversity, we minimize the Maximum Mean Discrepancy (MMD) between the selected subset and the full dataset, which is then reformulated as a Quadratic Unconstrained Binary Optimization (QUBO) problem. The resulting QUBO is submodular, which permits an efficient greedy algorithm. We further demonstrate feasibility on real quantum hardware through an end-to-end selection experiment. Our experimental results demonstrate that EBDAL not only improves the accuracy of uncertainty-based strategies but also outperforms a wide range of selection methods, achieving substantial computational speedups. The findings highlight EBDAL's robustness, efficiency, and adaptability across various datasets.

## 1 INTRODUCTION

Active learning (AL) is a learning paradigm in which a model interactively requests labels for the most informative samples from a large unlabeled pool, aiming to attain strong predictive performance with substantially fewer annotations. This is particularly relevant in modern deep learning, where unlabeled data are plentiful but labeling is costly, time-consuming, or requires domain expertise (e.g., medical imaging, autonomous driving). By selectively querying labels while maintaining accuracy, AL provides a practical route to scaling supervised learning under budget constraints. Methodologically, AL spans uncertainty sampling and Bayesian approximations (Lewis & Gale, 1994; Gal et al., 2017; Yoo & Kweon, 2019), geometric/coreset and representative selection (Sener & Savarese, 2018), gradient-based hybrid criteria (Ash et al., 2019), and variational/adversarial formulations (Sinha et al., 2019), with comprehensive surveys available in (Settles, 2009; Li et al., 2024). Collectively, these lines of work underscore the need for query strategies that jointly balance uncertainty, representativeness, and computational scalability.

Uncertainty-based active learning has achieved strong empirical success, reliably reducing annotation cost by prioritizing highly informative queries. Nevertheless, redundancy frequently arises under tight budgets: top-uncertainty samples often cluster within the same local region, leading to inefficient label usage (Figure 1). Recent efforts have begun to address these issues but leave important gaps. For example, Wang et al. (2024b) automates strategy choice from a set of heuristics, advancing ease of use; yet committing to a single policy per round limits the ability to balance uncertainty and diversity within a batch, and the strategy-search/preparation phases introduce nontrivial overhead that is often omitted from end-to-end timing. Likewise, Bae et al. (2025) improves representativeness via uncertainty-weighted coverage, but the approach remains sensitive to uncertainty calibration, is still prone to redundancy when uncertainty mass concentrates, and can under-sample globally representative regions with lower uncertainty. These observations suggest that a more effective solution should explicitly integrate uncertainty with distributional diversity while remaining computationally efficient across datasets and budget regimes.

To address these challenges, we propose Efficient Blockwise Diverse Active Learning (EBDAL), a scalable framework that couples uncertainty with distributional diversity. We operationalize diversity through maximum mean discrepancy (MMD), selecting a subset that approximates the kernel mean of the unlabeled pool, thereby promoting broad coverage and reducing redundancy. Instead of performing a single global optimization, EBDAL adopts a blockwise MMD minimization strategy, significantly improving time and memory efficiency. We establish a bounded approximation gap between the blockwise objective and the global optimum. To balance informativeness and representativeness, our method selects samples from a high-uncertainty candidate set without over-relying on the precise uncertainty values. Technically, the blockwise MMD selection is formulated as a Quadratic Unconstrained Binary Optimization (QUBO) problem with a submodular objective. This formulation admits an efficient greedy algorithm in practice and is also compatible with modern quantum hardware.

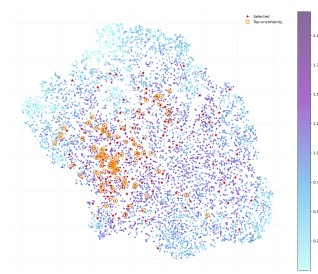

Figure 1: Visualization example of redundancy

**The main contributions in this work are as follows:**

- **A scalable, redundancy-aware AL framework.** We introduce **EBDAL**, which integrates uncertainty with *distributional diversity* through *blockwise* MMD minimization. Operating per block promotes broad feature-space coverage and substantially reduces small-batch redundancy common to uncertainty-only selection, while remaining simple to deploy on large unlabeled pools.

- **Theory and efficiency.** We provide a *bounded approximation guarantee* showing that blockwise MMD closely tracks the global MMD objective, preserving distributional alignment. In practice, our complexity analysis and wall-clock experiments show consistent speedups compared with full-pool, without loss of accuracy.

- **Quantum-enabled selection.** We formulate the blockwise MMD selection as a QUBO problem whose objective is submodular. This formulation enables two complementary solving approaches: (i) a fast greedy algorithm for practical applications, and (ii) direct access to real quantum hardware. To our knowledge, this yields the first demonstration of an end-to-end active-learning loop that uses a quantum computer for the selection phase and a classical computer for model training, applied to classical datasets.

## 2 RELATED WORK

**Active Learning** Active learning (AL) targets high performance under tight labeling budgets by querying the most informative points. Early work formalized uncertainty sampling for text (Lewis & Gale, 1994). With deep models, approximate Bayesian inference enabled mutual-information criteria such as Bayesian Active Learning by Disagreement (Gal et al., 2017); learnable acquisition functions predict per-sample loss (Yoo & Kweon, 2019); and batch-diverse methods curb redundancy via gradient-space clustering (BADGE) (Ash et al., 2019) or variational adversarial latent-space selection (Sinha et al., 2019). Deterministic uncertainty estimation (DDU) provides a strong non-Bayesian baseline (Mukhoti et al., 2023). Recent efforts emphasize scalability and automation, including leverage-score sampling (Shimizu et al., 2023), uncertainty herding (Bae et al., 2025), and differentiable strategy search (Wang et al., 2024b).

**Coreset Selection** Coresets approximate large datasets with small, weighted summaries to accelerate training while maintaining accuracy. In AL, geometry-driven $k$-center selection reduces redundancy for CNNs (Sener & Savarese, 2018). More broadly, coreset techniques offer scalable summaries with provable guarantees (Mirzasoleiman et al., 2020) and bilevel updates for streaming and continual learning (Borsos et al., 2020). Recent diversity-centric subset design revisits coverage for robustness (e.g., instruction tuning) (Wang et al., 2024a). Our approach is coreset-like in spirit but differs by optimizing an RKHS-grounded diversity objective (MMD) in a blockwise, active pipeline that includes uncertainty filtering and an acceleration path via a QUBO reformulation.

**Submodularity and QUBO/Ising** For subset selection under a cardinality constraint, constructing (approximately) submodular objectives is a powerful design principle. Scalable variants tailor this idea to machine learning, e.g., Bayesian batch AL as sparse subset approximation (Pinsler et al., 2019), and distributed optimization for pairwise submodular functions (Böther et al., 2024). A complementary line formulates selection as QUBO/Ising (see modeling and mapping references (Lucas, 2014; Kochenberger et al., 2014; Glover et al., 2018)), enabling specialized solvers and quantum optimizers such as quantum annealing (Kadowaki & Nishimori, 1998) and QAOA (Farhi et al., 2014). In contrast to (Bae et al., 2025), which relies on uncertainty-weighted coverage and can be sensitive to miscalibration and cluster concentration, our method optimizes an explicit MMD-based distributional alignment with blockwise structure and a provable approximation to the global objective, yielding stronger redundancy mitigation and robustness.

## 3 METHODOLOGY

### 3.1 PROBLEM DEFINITION

We consider a standard pool-based active learning setup. Let $\mathcal{D}_U = \{\mathbf{x}_i\}_{i=1}^N$ denote a large pool of unlabeled data instances. The learning process begins with a small labeled set $\mathcal{D}_L = \{(\mathbf{x}_j, y_j)\}_{j=1}^M$, where $M \ll N$ and $y_j$ represents the ground-truth label for $\mathbf{x}_j$. The core of the problem is a sequential querying procedure governed by a fixed annotation budget $Q$. In each iteration $t$, an acquisition function $\alpha(\mathbf{x}, \mathcal{M}_{t-1})$, which quantifies the utility of querying the label of a point $\mathbf{x}$, is used to select a batch of $b$ samples from the unlabeled pool:

$$\mathcal{D}_q^t = \operatorname*{argmax}_{\mathcal{D} \subset \mathcal{D}_U, |\mathcal{D}|=b} \sum_{\mathbf{x} \in \mathcal{D}} \alpha(\mathbf{x}, \mathcal{M}_{t-1}), \tag{1}$$

The oracle then provides the labels $\{y\}$ for the queried batch $\mathcal{D}_q^t$. These newly labeled samples are removed from $\mathcal{D}_U$ and added to $\mathcal{D}_L$. The model $\mathcal{M}_{t-1}$ is then trained on the newly labeled batch $\mathcal{D}_q^t$, and the training cycle repeats. This select-query-retrain cycle continues until the annotation budget $Q$ is exhausted. The objective is to maximize the sample efficiency of the learning process, obtaining a highly accurate model with as few labeled examples as possible.

To mitigate the redundancy of conventional global top-$b$ uncertainty sampling, we adopt a filter–then–match paradigm that couples uncertainty with diversity. We first retain a generous set of high-uncertainty candidates, yielding a filtered search pool

$$\mathcal{D}_U^{\mathrm{flt},t} = \{\, x \in \mathcal{D}_U \mid x \text{ is among the } k \text{ most uncertain points at iteration } t \,\}, \qquad k \gg b. \tag{2}$$

This step uses uncertainty purely as a screen: it prunes clearly low-utility points and curbs local redundancy, while preserving a high-recall set of plausible candidates. Given the screened pool, the final batch is selected by aligning with the distribution of the entire unlabeled set via MMD:

$$\mathcal{D}_q^t = \operatorname*{argmin}_{\mathcal{D} \subset \mathcal{D}_U^{\mathrm{flt},t}, |\mathcal{D}|=b} \mathrm{MMD}^2(\mathcal{D}, \mathcal{D}_U). \tag{3}$$

Uncertainty forms a high-recall candidate set, and MMD enforces coverage/representativeness in the final picks—yielding batches that are both informative and diverse. To scale to large datasets, we first cluster the pool into feature-space blocks and then apply Eq.(2) and Eq.(3) in parallel within each block before aggregating the selections.

### 3.2 DIVERSITY-ENHANCED

Active learning under a fixed budget can be viewed as iterative coreset selection: at each round, we choose a small subset that minimizes the expected training loss (i.e., the loss gap relative to using the full pool). This perspective turns batch acquisition into selecting representative points that preserve loss, naturally motivating the joint use of uncertainty and coverage/diversity. Formally, the objective can be expressed as:

$$\min_{\mathcal{D}_q^t := b} \mathbb{E}_{x, y \sim P}\left[l(x, y; \mathcal{M}_t)\right], \tag{4}$$

where $\mathbf{x}_i, y_i$ are i.i.d. drawn from an underlying distribution $P$, and $l(x, y; \mathcal{M}_{t-1})$ is the loss function of the model $\mathcal{M}_{t-1}$ after being trained on the batch $\mathcal{D}_q^t$.

Directly minimizing the training loss in Eq. (4) is computationally challenging. To overcome this, we introduce a tractable surrogate objective: instead of minimizing the loss directly, we propose to minimize the MMD between the selected subset and the full dataset. We establish a connection between the upper and lower bounds of the training loss function over a given subset and the MMD of that subset with respect to the full unlabeled pool. Specifically, it measures the distance between the distribution of the selected coreset and that of the full unlabeled set. The formal statement of MMD between two distributions $P$ and $Q$ is as follows:

$$\text{MMD}[\mathcal{F}, P, Q] = \sup_{\|f\|_{\mathcal{H}} \leq 1} \left( \mathbb{E}_P[f(x)] - \mathbb{E}_Q[f(y)] \right), \tag{5}$$

where $\mathcal{F}$ is a class of functions, and $\|f\|_{\mathcal{H}}$ denotes the norm of the function $f$ in the reproducing kernel Hilbert space (RKHS). The expectations $\mathbb{E}_P[f(x)]$ and $\mathbb{E}_Q[f(y)]$ represent the expected values of the function $f$ under the distributions $P$ and $Q$. The MMD measures the largest difference between the distributions $P$ and $Q$ in terms of the function class $\mathcal{F}$. By minimizing the MMD, we encourage the selected subset to closely match the full distribution of the unlabeled data, ensuring that the selected coreset is a good approximation of the entire dataset. Before formally establishing the connection between the MMD discrepancy and the expected loss, we first state the following key lemma.

**Lemma 1.** *(Zheng et al., 2022) Given $n$ i.i.d. samples drawn from $P_\mu$ as $S = \{(x_i, y_i)\}_{i \in [n]}$, where $y_i \in [C]$ is the class label for example $x_i$, a coreset $S'$ that is a $p$-partial $r$-cover for $P_\mu$ on the input space $X$, and any $\varepsilon > 1 - p$, suppose: (i) the loss $l(\cdot, y, w)$ is $\lambda_\ell$-Lipschitz continuous for all $y, w$ and bounded by $L$; (ii) the class-specific regression function $\eta_c(x) := \mathbb{P}(y = c \mid x)$ is $\lambda_\eta$-Lipschitz for all $c \in [C]$; and (iii) $l(x, y; h_{S'}) = 0$ for all $(x, y) \in S'$. Then, with probability at least $1 - \varepsilon$,*

$$\left| \frac{1}{n} \sum_{(x,y) \in S} l(x, y; h_{S'}) \right| \leq r\left(\lambda_\ell + \lambda_\eta LC\right) + L\sqrt{\frac{\log\left(\frac{p}{p + \varepsilon - 1}\right)}{2n}}. \tag{6}$$

Informally, we interpret a $p$-partial $r$-cover as the fraction $p$ of points in the ground set $X$ that fall within distance $r$ of the subset $S$ (the formal definition is deferred to the Appendix A). By the preceding lemma, with probability at least $p$, the loss in a single training pass is upper-bounded by a quantity that is monotone in the covering radius $r$, and therefore is primarily determined by $r$. Hence minimizing the loss can be viewed as selecting points with strong covering power under the $p$-partial $r$-cover criterion. To facilitate such selection, we relate coverage to the empirical MMD below.

**Theorem 1** (Coverage–MMD relation). *Let $X$ be the full dataset and $S \subseteq X$ a subset. For a radius $r > 0$, denote the coverage ratio by*

$$\rho(r) := \frac{1}{|X|} \left| \{ x \in X : \text{dist}(x, S) \leq r \} \right|.$$

*Let $k(\cdot, \cdot)$ be the Gaussian kernel $k(u, v) = \exp\left(-\|u - v\|^2/(2\sigma^2)\right)$. Then the following bounds hold:*

$$\frac{\delta_{\text{MMD}} - e^{-r^2/(2\sigma^2)}}{1 - e^{-r^2/(2\sigma^2)}} < \rho(r) < \delta_{\text{MMD}} |S| e^{r^2/(2\sigma^2)}. \tag{7}$$

*Here $\delta_{\text{MMD}}$ denotes the cross term in the empirical MMD between $S$ and $X$.*

Theorem 1 establishes a precise connection between the MMD term $\delta_{\text{MMD}}$ and data coverage. The bounds in Eq.(7) show that a larger $\delta_{\text{MMD}}$ (implying a smaller MMD) tightens both the lower and upper bounds on the coverage ratio $\rho(r)$ for any radius $r$. This means that for a fixed target coverage level $p$, the required radius $r_p$ decreases as $\delta_{\text{MMD}}$ increases. Combining this with Lemma 1 yields the key insight: minimizing the MMD leads to a tighter probabilistic loss bound for a given success probability $p$, by reducing the effective covering radius needed.

## 3.3 FROM MMD-BASED DIVERSITY TO A QUBO FORMULATION

When $P$ and $Q$ are represented by empirical samples $X = \{x_i\}_{i=1}^n$ and $S \subseteq X$, and using the kernel mean embeddings $\mu_X = \frac{1}{n} \sum_{i=1}^n \phi(x_i)$ and $\mu_S = \frac{1}{|S|} \sum_{x \in S} \phi(x)$, the squared MMD admits a closed-form expression:

$$f(S) = \left\| \mu_S - \mu_X \right\|_{\mathcal{H}}^2.$$

Let $K \in \mathbb{R}^{n \times n}$ be the Gram matrix with $K_{ij} = \phi(x_i) \cdot \phi(x_j) = k(x_i, x_j)$ and $m \in \{0, 1\}^n$ encode the selected subset with $\mathbf{1}^\top m = k$. The MMD-minimization problem is

$$\min_{m \in \{0,1\}^n} \frac{1}{k^2} m^\top K m - \frac{2}{nk} \mathbf{1}^\top K m \quad \text{s.t.} \quad \mathbf{1}^\top m = k. \tag{8}$$

With a sufficiently large penalty on the cardinality constraint, Eq.(8) is directly converted into a QUBO of the form $\min_{m \in 0,1^n} m^\top \widehat{Q}, m$ having the same minimizers; the construction of $\widehat{Q}$ and the equivalence proof are deferred to Appendix B.Moreover, since common kernels satisfy $K_{ij} \geq 0$, maximizing the negated objective of Eq.(8) under the same cardinality constraint yields a quadratic set function whose continuous relaxation has the Hessian with non-positive off-diagonal entries, hence it is submodular (Bian et al., 2017). Therefore, the instance can be tackled either by (cardinality-constrained) greedy selection or by quantum QUBO/Ising solvers (see Appendix B).

Even with extensive engineering, greedy selection over the full pool is costly. If we precompute the non-sparse Gram matrix once and update marginal gains incrementally, the computational complexity of $k$-step greedy procedure on $n$ points is as follows:

$$T_{\text{greedy}}^{\text{full}} = O(n^2 + kn). \tag{9}$$

After partitioning $X = \bigcup_{b=1}^B X_b$ with $|X_b| = n_b$ and running greedy *inside each block*, the total cost becomes

$$T_{\text{greedy}}^{\text{block}} = O\left( \sum_{b=1}^B n_b^2 + \sum_{b=1}^B k_b n_b \right), \qquad \sum_{b=1}^B k_b = k. \tag{10}$$

When blocks are reasonably balanced ($n_b \approx n/B$ and $k_b \approx k/B$),

$$T_{\text{greedy}}^{\text{block}} = O\left( \frac{n^2}{B} + \frac{kn}{B^2} \right), \tag{11}$$

the QUBO in Eq.(8) involves one binary variable per data point, making it prohibitively slow and yielding poor solutions on large datasets for both classical and quantum solvers. Our blockwise partitioning confines each QUBO instance to a scale of a few hundred variables, which is well within the regime where classical optimizers perform efficiently and effectively, and is also compatible with the capabilities of current quantum hardware.

To strengthen the alignment between a small MMD and strong coverage, as suggested by Theorem 1 which becomes tighter for data blocks with smaller diameter, we partition the data into compact clusters (e.g., in kernel feature space) using K-means for its simplicity and scalability. Moreover, this blockwise approach is principled: when the selection quota for each block is set proportional to its size, optimizing MMD separately within each block serves as a surrogate for the global objective, since the latter is governed by a weighted average of the per-block objectives (a generalized statement with a quota-mismatch penalty is given in Appendix C).

**Theorem 2** (Blockwise MMD approximation with proportional quotas). *Let $X = \bigcup_{b=1}^B X_b$ be a partition with weights $w_b := |X_b|/|X|$. From each block $b$, select $S_b \subseteq X_b$ of size $|S_b| = k_b$ and set $S = \bigcup_{b=1}^B S_b$, with $\alpha_b := k_b/k$ and $\sum_{b=1}^B \alpha_b = \sum_{b=1}^B w_b = 1$. Let $k$ be a bounded positive definite kernel with feature map $\phi$ into an RKHS $\mathcal{H}$, and assume $\sup_x k(x,x) \leq \kappa^2$ (for a normalized RBF kernel, $\kappa = 1$). Denote $\mu_A := |A|^{-1} \sum_{x \in A} \phi(x)$ and $\mathrm{MMD}^2(A, B) := \|\mu_A - \mu_B\|_{\mathcal{H}}^2$. If quotas are proportional, i.e., $\alpha = w$, then*

$$\mathrm{MMD}^2(S, X) \leq 2 \sum_{b=1}^B w_b \, \mathrm{MMD}^2(S_b, X_b) \leq 2 \max_b \mathrm{MMD}^2(S_b, X_b). \tag{12}$$

*In particular, if each blockwise procedure achieves $\mathrm{MMD}^2(S_b, X_b) \leq \varepsilon_b$, then $\mathrm{MMD}^2(S, X) \leq 2 \sum_b w_b \varepsilon_b \leq 2 \max_b \varepsilon_b$.*

Theorem 2 indicates that with proportional quota allocation ($\alpha = w$), the global MMD error is bounded by a weighted average of the per-block errors. Thus, by ensuring the selection from each compact cluster is representative, the blockwise procedure closely approximates the global objective. Consequently, optimizing MMD within each block is a principled surrogate for the intractable global optimization. If the quotas deviate from this proportion, an additional penalty term is incurred.

### 3.4 ALGORITHM

Given a candidate set $C_b \subseteq X_b$ and quota $k_b$, we solve the per-block MMD problem

$$\min_{S_b \subseteq C_b,\, |S_b|=k_b} \text{MMD}^2(S_b, X_b) = \left\| \frac{1}{k_b} \sum_{x \in S_b} \phi(x) - \frac{1}{n_b} \sum_{x \in X_b} \phi(x) \right\|_{\mathcal{H}}^2. \tag{13}$$

*With proportional quotas ($k_b \propto |X_b|$), Theorem 2 ensures that the global error is controlled by a weighted average of the per-block errors.*

---

**Algorithm 1** EBDAL (multi-round active learning)

---

**Require:** Initial labeled set $L_0$, unlabeled pool $U_0$, number of AL rounds $T$, per-round batch size $k$, candidate ratio $\tau$, max block size $M_{\max}$, kernel $k(\cdot, \cdot)$, uncertainty scorer $u_\theta(\cdot)$, SOLVER for Eq. equation 13

**Ensure:** Final trained model $\theta_T$

1: $L \leftarrow L_0, U \leftarrow U_0$
2: **for** $t = 1$ to $T$ **do**                                                   ▷ outer AL loop
3:       Train / fine-tune model parameters $\theta$ on $L$
4:       Compute embeddings $z_\theta(x)$ for all $x \in U$
5:       Cluster $U$ in the kernel/feature space (e.g., recursive Kmeans) so each block size $\leq M_{\max}$: $U = \bigcup_{b=1}^{B} X_b, n_b \leftarrow |X_b|$
6:       Set block quotas: $k_b \leftarrow \text{round}\big(k\, n_b/|U|\big)$; adjust by $\pm 1$ to ensure $\sum_b k_b = k$
7:       Set global candidate budget $M \leftarrow \lceil \tau |U| \rceil$ and distribute to blocks: $c_b \leftarrow \min\{ n_b, \max(k_b, \text{round}(M\, n_b/|U|)) \}$; adjust by $\pm 1$ so $\sum_b c_b = M$
8:       $S^{(t)} \leftarrow \varnothing$
9:       **for** $b = 1$ to $B$ **do**
10:          Rank $X_b$ by uncertainty $u_\theta(\cdot)$ (descending); let $C_b$ be the top-$c_b$ in $X_b$
11:          (Optional) Fit per-block kernel parameters (e.g., RBF bandwidth via the median heuristic on $X_b$)
12:          **Solve** the per-block objective: $S_b \leftarrow \text{SOLVER}(C_b, X_b, k_b;\ k(\cdot, \cdot))$ for Eq. (13)
13:          $S^{(t)} \leftarrow S^{(t)} \cup S_b$
14:       Query labels $y$ for all $x \in S^{(t)}$ from the oracle
15:       $L \leftarrow L \cup \{(x, y) : x \in S^{(t)}\}, \qquad U \leftarrow U \setminus S^{(t)}$
16:       Train model $f_{\theta_T}$ using $\{(x, y) : x \in S^{(t)}\}$
17: **return** $\theta_T$

---

**Remarks.** (i) By setting quotas proportional to block sizes ($k_b \propto n_b$), we adhere to the conditions of Theorem 2, which eliminates the quota-mismatch penalty and bounds the global MMD by a weighted sum of the local block MMDs.

(ii) Candidate filtering ($c_b \ll n_b$) reduces the kernel computation cost within each block from $O(n_b^2)$ to $O(c_b^2)$ and significantly shrinks the problem size for the downstream solver.

(iii) Blocks are processed independently, enabling parallelization. The framework is solver-agnostic, accommodating various backends (e.g., QUBO/Ising solvers; see Appendix D).

(iv) **Hardware-aware Ising mapping.** To cope with the limited bitwidth and dynamic range of current quantum hardware, we adopt a precision-aware multi-auxiliary construction that *splits large data-spin fields* into several couplers to a small set of ancilla spins. This redistributes scale so that individual coefficients are comparable to the typical $J_{ij}$ level after global rescaling/quantization, thereby preserving the effective resolution of $J$ and mitigating precision loss; implementation choices (e.g., $\Delta_J, \gamma, G_i, B_g$) are detailed in Appendix D.

## 4 EXPERIMENTS

We evaluate our diversity-enhanced acquisition framework on standard image-classification benchmarks by integrating it with representative uncertainty criteria and comparing each baseline to its diversity-enhanced counterpart, reporting both accuracy-budget curves and the Area Under the Curve (AUC). We also benchmark the selection backend by comparing a fast greedy solver against a quantum QUBO/Ising formulation. Finally, we perform ablations that successively enable three core modules: block partitioning with proportional per-block budgets; an in-block uncertainty filter that forms a candidate pool; and MMD-based selection within each block. This setup isolates their individual contributions.

### 4.1 EXPERIMENT SETTINGS

**Datasets and Baselines** We evaluate on five image classification benchmarks: CIFAR-10, CIFAR-100 (Krizhevsky et al., 2009), MNIST (Deng, 2012), SVHN (Netzer et al., 2011), and Fashion-MNIST (Xiao et al., 2017). To study the benefit of diversity, we combine our framework with three representative uncertainty-based selectors: *entropy* (probability-based) (Shannon, 2001), *BALD* (Bayesian active learning by disagreement) (Gal et al., 2017), and *LPL* (learning-loss predictor) (Yoo & Kweon, 2019). Methods suffixed with "`-EBDAL`" denote the corresponding algorithms augmented by our diversity module.

As baselines, we include the standalone versions of the above three uncertainty methods (*entropy*, *BALD*, *LPL*), as well as *BADGE* (Ash et al., 2019), *Kmeans* (Lloyd, 1982) clustering–based selection, and the recent uncertainty-herding approach *Uherding* (Bae et al., 2025). Further implementation details (initial labeled size, labeling budgets, model/backbone, training schedules, and hyperparameters) are provided in the Appendix E.

### 4.2 PERFORMANCE COMPARISON ACROSS DATASETS

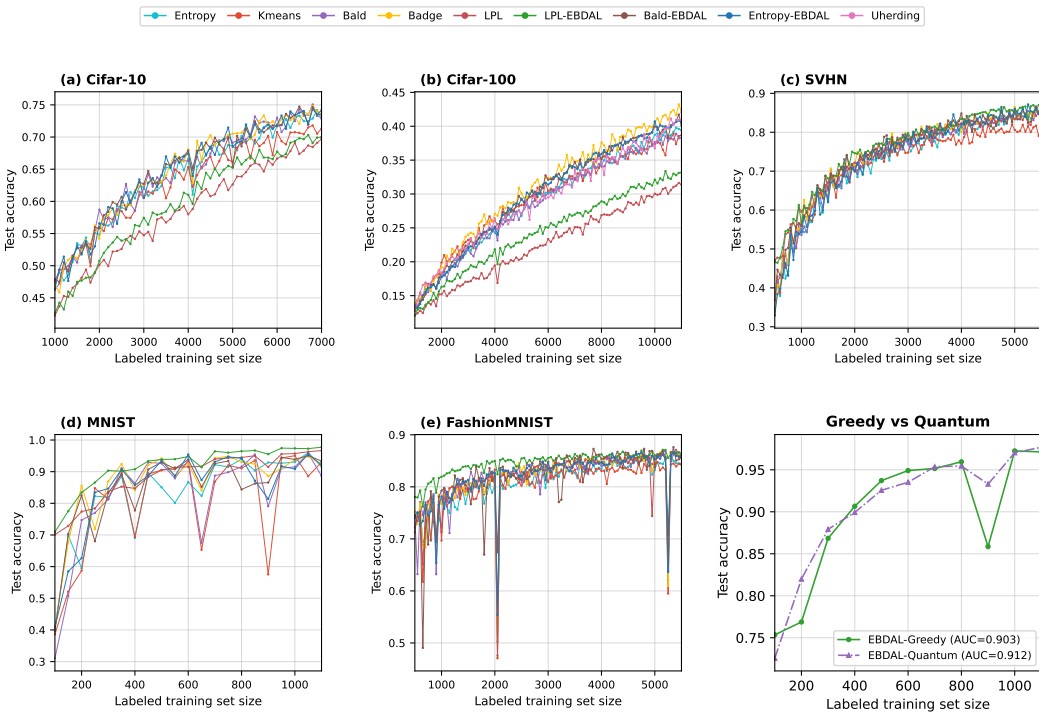

Figure 2: **Overall performance trends across labeling budgets.** Test accuracy vs. labeled training set size on five benchmarks.

| Method | CIFAR-10 | CIFAR-100 | SVHN | MNIST | FMNIST |
|---|---|---|---|---|---|
| Bald | 0.6450 | 0.2866 | 0.7351 | 0.8392 | 0.8110 |
| Bald-EBDAL | **0.6482 (+0.32)** | 0.3005 (+1.39) | 0.7440 (+0.89) | 0.8589 (+1.97) | 0.8180 (+0.70) |
| Entropy | 0.6391 | 0.2902 | 0.7259 | 0.8418 | 0.8061 |
| Entropy-EBDAL | 0.6469 (+0.78) | 0.2997 (+0.95) | 0.7359 (+1.00) | 0.8573 (+1.55) | 0.8212 (+1.51) |
| LPL | 0.5851 | 0.2258 | 0.7478 | 0.8880 | 0.8184 |
| LPL-EBDAL | 0.5977 (+1.26) | 0.2446 (+1.88) | **0.7595 (+1.17)** | **0.9219 (+3.39)** | **0.8467 (+2.83)** |
| Badge | 0.6480 | **0.3107** | 0.7448 | 0.8748 | 0.8188 |
| Kmeans | 0.6282 | 0.2940 | 0.7209 | 0.8146 | 0.8039 |
| Uherding | 0.6441 | 0.2923 | / | / | / |

Table 1: AUC (Area Under Curve) across datasets. Best per dataset in **bold**. Values in parentheses show the improvement over the corresponding non-EBDAL baseline (in percentage points).

Across the five image classification benchmarks, we observe a consistent pattern: for a fixed labeling budget, uncertainty methods augmented with our diversity module (the "-EBDAL" variants) shift the entire accuracy–budget curve upward relative to their uncertainty-only counterparts. This translates into fewer redundant queries and more effective use of labels (see Figure 2; quantitative AUCs appear in Table 1).

The gains are especially pronounced on settings with many classes and noisier raw uncertainty signals (e.g., CIFAR-100), where both *LPL-EBDAL* and *Entropy-EBDAL* deliver clear AUC improvements over their baselines. On easier datasets such as MNIST and SVHN, where most methods already perform strongly at small budgets, EBDAL variants still yield consistent positive shifts; notably, *LPL-EBDAL* attains the best AUC on MNIST.

Compared with diversity-only baselines (e.g., *KMeans*, *BADGE*), EBDAL achieves a more robust exploration–exploitation balance by first forming a high-uncertainty candidate pool within each block and then selecting a subset that minimizes MMD to the block population. This design avoids small-batch redundancy caused by concentrated uncertainty mass while preserving global representativeness and scalability.

The sixth panel further contrasts greedy and quantum backends under the same setup. The two curves closely align, with AUCs of 0.903 (*EBDAL-Greedy*) and 0.912 (*EBDAL-Quantum*), demonstrating the feasibility of an end-to-end selection–training loop on real hardware. In our testbed, the quantum backend solves a 500-variable QUBO instance in about $0.2\,\text{ms}$ per call, which is orders of magnitude faster than the second-scale runtime of the greedy (submodular) solver on problems of comparable size. Given the current invocation associated with quantum resources, we therefore use the greedy solver for the main experiments and ablations, and report the quantum results as a feasibility and effectiveness reference. A representative Hamiltonian-energy evolution trace and brief explanation are provided at the end of Appendix E.

Overall, EBDAL delivers consistent accuracy gains and strong computational scalability across datasets and budgets, validating the benefit of integrating blockwise diversity with uncertainty in practical active learning pipelines.

### 4.3 ABLATION STUDIES

We ablate the three components of our diversity-enhanced acquisition pipeline—(i) block partitioning in feature space, (ii) per-block uncertainty filtering that forms an uncertainty candidate pool, and (iii) distributional selection via in-block MMD minimization—using three variants: *LPL (standard)* without blocks or explicit diversity; *Block-TopU* with block partitioning and UCPs but selecting the $k_b$ most-uncertain points per block; and *LPL-EBDAL* with the full blockwise MMD selection. All configurations share the same backbone, training schedule, data splits, and label budgets; curves report test accuracy versus labeled-set size to isolate acquisition effects.

Across MNIST, CIFAR-10, and SVHN, the full *LPL-EBDAL* consistently dominates its ablations over a wide range of budgets, indicating that explicit distributional matching (MMD) adds value beyond uncertainty guidance and block partitioning alone. *Block-TopU* improves over *LPL* in the low-budget regime by dispersing queries across blocks (reducing obvious redundancy), but it tends

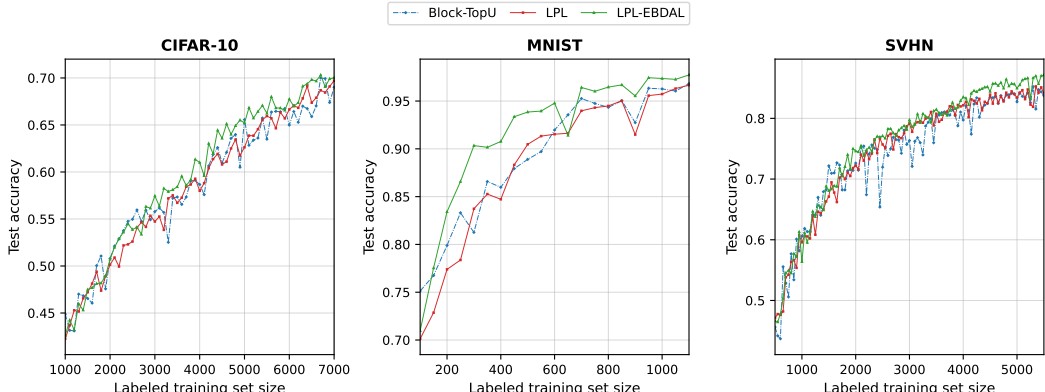

Figure 3: Ablation on *MNIST*, *CIFAR10*, and *SVHN*. We compare **LPL** (no blocking/diversity), **Block-TopU** (blocking + per-block top-uncertainty), and **LPL-EBDAL** (full). **LPL-EBDAL** consistently achieves the best label-efficiency by combining uncertainty guidance with MMD-based distributional matching within each block.

to plateau earlier when high-uncertainty points cluster within the same local regions. In contrast, *LPL-EBDAL* continues to benefit as the budget grows, suggesting that aligning the selected subset's kernel mean with each block's population mitigates within-block redundancy and recovers globally representative samples that pure uncertainty may undervalue. Representative trajectories are shown in Figure 3.

## 5 CONCLUSION

We presented Efficient Blockwise Diverse Active Learning (EBDAL), a practical framework that integrates uncertainty guidance with distributional diversity through blockwise MMD minimization. By framing active learning as iterative coreset selection, we establish a theoretical link between distributional alignment and loss control. Technically, we formulate the per block MMD objective as a submodular QUBO, enabling efficient greedy selection and compatibility with quantum solvers. Theoretically, we derive a coverage MMD bound and prove that blockwise optimization with proportional quotas closely approximates the global objective. Empirically, EBDAL consistently improves accuracy budget curves and AUC over uncertainty only and diversity only baselines across standard image benchmarks, while reducing runtime. Ablations validate the contribution of each component, and a direct comparison demonstrates the feasibility of executing the selection phase on real quantum hardware.

In summary, EBDAL delivers a scalable and label efficient approach to active learning by coupling uncertainty with principled blockwise distribution matching, providing a practical bridge to advanced optimizers without sacrificing accuracy.

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
