LARGE MODEL USAGE STATEMENT

We used large language models for text translation and polishing.

## A PROOF OF THEOREM1

In this section we make precise the notion of a $p$-partial $r$-cover and present a complete proof of Theorem 1. We first recall the formal definition of $p$-partial $r$-cover. We then prove two-sided bounds that relate $\rho(r)$ to the empirical MMD cross term under a Gaussian kernel. This establishes a rigorous connection between RKHS-based distributional discrepancy and geometric coverage, clarifying why minimizing MMD promotes stronger coverage.

**Definition 1** ($p$-partial $r$-cover). *Following (Zheng et al., 2022), given a metric space $(X, d)$ and a probability measure $\mu$ on $X$ (with density $P_\mu$), a set $S \subset X$ is a p-partial r-cover if*

$$\int_X \mathbf{1}_{\bigcup_{x \in S} B_d(x,r)}(x)\, d\mu(x) \;=\; p,$$

*where $B_d(x, r) = \{\, x' \in X : d(x, x') \leq r \,\}$ is the radius-r ball centered at $x$, and $\mathbf{1}_A$ denotes the indicator of a set $A$.*

*Proof.* Let

$$A := \{\, x \in X : \operatorname{dist}(x, S) \leq r \,\}, \qquad |A| = |X|\, \rho(r). \tag{14}$$

Write the (standard) MMD cross average as

$$\delta_{\mathrm{MMD}} \;=\; \frac{1}{|S|\,|X|} \sum_{s \in S} \sum_{x \in X} k(s, x) \;=\; \frac{1}{|X|} \sum_{x \in X} \Big( \frac{1}{|S|} \sum_{s \in S} k(s, x) \Big). \tag{15}$$

For any $x \in A$ there exists $s^\star \in S$ with $\|x - s^\star\| \leq r$, hence

$$\frac{1}{|S|} \sum_{s \in S} k(s, x) \;\geq\; \frac{1}{|S|} k(s^\star, x) \;\geq\; \frac{1}{|S|}\, e^{-\,r^2/(2\sigma^2)}. \tag{16}$$

Averaging Eq.(16) over $x \in A$ and discarding the (nonnegative) contribution from $X \setminus A$ gives

$$\delta_{\mathrm{MMD}} \;\geq\; \frac{|A|}{|X|} \cdot \frac{1}{|S|}\, e^{-\,r^2/(2\sigma^2)} \;=\; \frac{\rho(r)}{|S|}\, e^{-\,r^2/(2\sigma^2)}. \tag{17}$$

Therefore

$$\rho(r) \;\leq\; \delta_{\mathrm{MMD}}\, |S|\, e^{\,r^2/(2\sigma^2)}. \tag{18}$$

For any $x \notin A$ we have $\operatorname{dist}(x, S) > r$, so $k(s, x) \leq e^{-\,r^2/(2\sigma^2)}$ for all $s \in S$, while $k(s, x) \leq 1$ always. Hence

$$\frac{1}{|S|} \sum_{s \in S} k(s, x) \;\leq\; \begin{cases} 1, & x \in A, \\ e^{-\,r^2/(2\sigma^2)}, & x \notin A. \end{cases} \tag{19}$$

Averaging Eq.(19) over $x \in X$ yields

$$\delta_{\mathrm{MMD}} \;\leq\; \frac{1}{|X|} \Big( |A| \cdot 1 \,+\, (|X| - |A|)\, e^{-\,r^2/(2\sigma^2)} \Big) \;=\; \rho(r) \,+\, \big(1 - \rho(r)\big)\, e^{-\,r^2/(2\sigma^2)}. \tag{20}$$

Solving Eq.(20) for $\rho(r)$ gives

$$\rho(r) \;\geq\; \frac{\delta_{\mathrm{MMD}} - e^{-\,r^2/(2\sigma^2)}}{1 - e^{-\,r^2/(2\sigma^2)}}. \tag{21}$$

Combining Eq.(18) and Eq.(21) proves Eq.(7). $\qquad\square$

## B  FROM MMD MINIMIZATION TO QUBO

We briefly show how the cardinality–constrained MMD objective becomes a single-matrix QUBO. From Eq.(22), (i) add the quadratic penalty $\lambda(\mathbf{1}^\top m - k)^2$ to enforce $\mathbf{1}^\top m = k$, and (ii) use the binary identity $m_i^2 = m_i$ to absorb the linear term into the diagonal, yielding a standard QUBO with matrix $\widehat{Q}$. For sufficiently large $\lambda$, minimizers coincide. We also give element-wise coefficients of $\widehat{Q}$ for implementation.

Let $K \in \mathbb{R}^{n \times n}$ be the Gram matrix with $K_{ij} = k(x_i, x_j)$ and $m \in \{0,1\}^n$ encode the selected subset with $\mathbf{1}^\top m = k$. The empirical MMD objective between the subset and the full pool reduces to

$$\min_{m \in \{0,1\}^n} \underbrace{\frac{1}{k^2} m^\top K m}_{\text{quadratic}} - \underbrace{\frac{2}{nk} \mathbf{1}^\top K m}_{\text{linear}} \quad \text{s.t.} \quad \mathbf{1}^\top m = k. \tag{22}$$

**Step 1 (penalize the cardinality constraint).** Add a quadratic penalty with $\lambda > 0$:

$$\lambda\big(\mathbf{1}^\top m - k\big)^2 = \lambda\, m^\top J m \; - \; 2k\lambda\, \mathbf{1}^\top m \; + \; \lambda k^2, \qquad J := \mathbf{1}\mathbf{1}^\top.$$

Dropping the constant $\lambda k^2$, we obtain the unconstrained binary quadratic objective

$$\min_{m \in \{0,1\}^n} \; m^\top Q m \; + \; q^\top m \; + \; \text{const}, \qquad Q := \frac{1}{k^2} K + \lambda J, \quad q := -\frac{2}{nk} K\mathbf{1} - 2k\lambda\mathbf{1}. \tag{23}$$

For sufficiently large $\lambda$, any minimizer of Eq.(23) satisfies $\mathbf{1}^\top m = k$ and thus solves Eq.(22).

**Step 2 (absorb the linear term into the quadratic).** For binary $m$, $m_i^2 = m_i$, hence $q^\top m = \sum_i q_i m_i = \sum_i q_i m_i^2 = m^\top \text{Diag}(q)\, m$. Define

$$\widehat{Q} \; := \; Q + \text{Diag}(q) \; = \; \frac{1}{k^2} K \; + \; \lambda J \; + \; \text{Diag}\!\Big(-\frac{2}{nk} K\mathbf{1} - 2k\lambda\mathbf{1}\Big).$$

Then we obtain a standard single-matrix QUBO:

$$\boxed{\min_{m \in \{0,1\}^n} \; m^\top \widehat{Q}\, m} \tag{24}$$

and Eq.(24) and Eq.(22) have the same minimizers (constants omitted). This QUBO can be handled by classical QUBO solvers or quantum Ising backends; the detailed equivalence proof and choices of $\lambda$ are in Appendix B.

**Element-wise coefficients of $\widehat{Q}$.** Let $u := K\mathbf{1}$. Then

$$\widehat{Q}_{ij} = \begin{cases} \dfrac{1}{k^2} K_{ij} + \lambda, & i \neq j, \\[2mm] \dfrac{1}{k^2} K_{ii} + \lambda \; - \; \dfrac{2}{nk} u_i \; - \; 2k\lambda, & i = j. \end{cases}$$

Here $J$ contributes $\lambda$ to *all* entries, including the diagonal; the linear term contributes only to the diagonal via $\text{Diag}(q)$.

## C  PROOF OF THEOREM 2

We bound the global RKHS discrepancy between the selected set and the pool by separating it into (a) within-block mismatches and (b) a quota-mismatch term that measures how far the selection weights $\alpha$ deviate from the block proportions $w$. Using a two-term inequality with a tunable constant, convexity, and bounded-kernel arguments, we obtain the blockwise approximation bound in Eq.( 31); when quotas are proportional ($\alpha = w$), it simplifies to Eq.( 32). The details follow.

We first note the blockwise decompositions

$$\mu_S = \sum_{b=1}^{B} \alpha_b\, \mu_{S_b}, \qquad \mu_X = \sum_{b=1}^{B} w_b\, \mu_{X_b}, \tag{25}$$

which imply

$$\mu_S - \mu_X = \underbrace{\sum_{b=1}^{B} \alpha_b \left( \mu_{S_b} - \mu_{X_b} \right)}_{\triangleq U} + \underbrace{\sum_{b=1}^{B} (\alpha_b - w_b) \, \mu_{X_b}}_{\triangleq V}. \tag{26}$$

**(i) Two-term split with a tunable constant.** For any $\eta > 0$, the parallelogram-type inequality

$$\|U + V\|^2 \ \leq \ (1+\eta)\|U\|^2 \ + \ \left(1 + \frac{1}{\eta}\right)\|V\|^2$$

holds. Taking $\eta = 1$ recovers the bound used in the main text:

$$\|\mu_S - \mu_X\|^2 \ \leq \ 2\|U\|^2 + 2\|V\|^2. \tag{27}$$

This shows where the factor 2 comes from and also allows a trade-off via $\eta$ when one term is known to be small.

**(ii) First term via Jensen/convexity.** Because $\sum_b \alpha_b = 1$ and $\|\cdot\|^2$ is convex,

$$\left\|\sum_{b=1}^{B} \alpha_b (\mu_{S_b} - \mu_{X_b})\right\|^2 \ \leq \ \sum_{b=1}^{B} \alpha_b \, \|\mu_{S_b} - \mu_{X_b}\|^2 \ = \ \sum_{b=1}^{B} \alpha_b \, \mathrm{MMD}^2(S_b, X_b). \tag{28}$$

**(iii) Second term via bounded kernels.** Assume $\sup_x k(x,x) \leq \kappa^2$. Then $\|\phi(x)\| \leq \kappa$ and hence

$$\|\mu_{X_b}\| = \left\|\frac{1}{|X_b|} \sum_{x \in X_b} \phi(x)\right\| \ \leq \ \frac{1}{|X_b|} \sum_{x \in X_b} \|\phi(x)\| \ \leq \ \kappa. \tag{29}$$

By the triangle inequality,

$$\left\|\sum_{b=1}^{B} (\alpha_b - w_b)\mu_{X_b}\right\| \ \leq \ \sum_{b=1}^{B} |\alpha_b - w_b| \, \|\mu_{X_b}\| \ \leq \ \kappa \, \|\alpha - w\|_1. \tag{30}$$

Combining Eq.(27)– Eq.(30) yields the general bound

$$\mathrm{MMD}^2(S, X) \ \leq \ 2\sum_{b=1}^{B} \alpha_b \, \mathrm{MMD}^2(S_b, X_b) \ + \ 2\kappa^2 \, \|\alpha - w\|_1^2. \tag{31}$$

**(iv) Proportional quotas as a corollary.** When $\alpha = w$ (quotas proportional to block sizes), the mismatch term vanishes and Eq.(31 )simplifies to

$$\mathrm{MMD}^2(S, X) \ \leq \ 2\sum_{b=1}^{B} w_b \, \mathrm{MMD}^2(S_b, X_b) \ \leq \ 2\max_b \mathrm{MMD}^2(S_b, X_b), \tag{32}$$

which is exactly the theorem stated in the merged main text.

**(v) Optional variants.** If additional structure is available, one may tighten the mismatch term:

- Using $\|\sum_b c_b \mu_{X_b}\| \leq \left(\sum_b c_b^2\right)^{1/2} \left(\sum_b \|\mu_{X_b}\|^2\right)^{1/2} \leq \kappa\sqrt{B} \, \|\alpha - w\|_2$ gives an alternative bound $2\kappa^2 B \, \|\alpha - w\|_2^2$, which can be sharper or looser than the $\ell_1$ bound depending on $B$ and the sparsity of $\alpha - w$.
- If blocks are very compact so that $\|\mu_{X_b} - \mu_X\| \leq \Delta$ for all $b$, then writing $V = \sum_b (\alpha_b - w_b)(\mu_{X_b} - \mu_X)$ yields $\|V\| \leq \Delta \, \|\alpha - w\|_1$, replacing $\kappa$ by the typically smaller dispersion $\Delta$.

**(vi) Equality and interpretation.** If $\alpha = w$ and each block attains $\mu_{S_b} = \mu_{X_b}$ (perfect local matching), then the right-hand side of Eq.(32) is 0, hence $\mathrm{MMD}^2(S, X) = 0$. Thus compact clustering plus proportional quotas makes the global error a convex combination of local errors, explaining why blockwise MMD optimization closely tracks the full-pool objective.

# D    Ising Model

This subsection has two parts. First, we recall the standard, closed-form mapping from a binary QUBO to an Ising model via the change of variables $x_i = (s_i + 1)/2$, yielding explicit expressions for the pairwise couplers $J_{ij}$, local fields $h_i$, and the constant shift. Second, we address a practical issue on current quantum/annealing hardware: coefficients are encoded with limited bitwidth and dynamic range. In many AL objectives the cardinality penalty aggregates into large local fields $h_i$ whose scale can dwarf the pairwise couplers $J_{ij}$; after global rescaling/quantization this compresses the effective resolution of $J_{ij}$. We therefore describe a *multi-auxiliary* scheme that redistributes overly large data-spin fields into several couplers to a small set of ancilla spins, so that each coupling sits on the same scale as typical $J_{ij}$. This improves numerical conditioning and retains more useful precision for $J$ in QUBO→Ising deployments.

## D.1    Standard QUBO to Ising mapping

Consider a QUBO in the upper-triangular form

$$\min_{x \in \{0,1\}^n} E_{\text{QUBO}}(x) = \sum_{i \le j} Q_{ij} \, x_i x_j + C_0, \tag{33}$$

where linear terms have been absorbed into the diagonal ($Q_{ii}$). Introduce Ising spins $s \in \{-1, 1\}^n$ via $x_i = (s_i + 1)/2$. Using

$$x_i x_j = \tfrac{1}{4} \left( 1 + s_i + s_j + s_i s_j \right) \quad (i < j), \qquad x_i = \tfrac{1}{2} \left( 1 + s_i \right),$$

we obtain an Ising Hamiltonian (up to an additive constant)

$$\min_{s \in \{-1,1\}^n} E_{\text{Ising}}(s) = -\sum_{i < j} J_{ij} \, s_i s_j - \sum_i h_i \, s_i + C, \tag{34}$$

with coefficients

$$J_{ij} = -\frac{Q_{ij}}{4} \quad (i < j), \qquad h_i = -\frac{Q_{ii}}{2} - \frac{1}{4} \sum_{j \ne i} Q_{ij}, \tag{35}$$

and constant $C = C_0 + \tfrac{1}{4} \sum_{i<j} Q_{ij} + \tfrac{1}{2} \sum_i Q_{ii}$ (which does not affect minimizers). Equations Eq.(33) and Eq.(34) therefore have the same minimizers after the change of variables $x_i = (s_i + 1)/2$.

**Multi-auxiliary (precision-aware) construction**    We start from the Ising form

$$E(s) = -\sum_{i < j} J_{ij} \, s_i s_j - \sum_{i=1}^{n} h_i \, s_i, \qquad s_i \in \{-1, 1\}. \tag{36}$$

When the bitwidth is limited, very large $|h_i|$ (e.g., from a cardinality penalty) dominate the global scale used for coefficient normalization/quantization, thereby squeezing the dynamic range available to represent $J_{ij}$. To alleviate this, we split each large field into several couplers to auxiliary spins so that every individual coupling magnitude is comparable to a chosen $J$-scale.

Concretely, introduce $G$ ancilla spins $a_g \in \{-1, 1\}$ and nonnegative weights $w_{ig}$ such that

$$\sum_{g=1}^{G_i} w_{ig} = |h_i|, \qquad w_{ig} \approx \Delta_J, \tag{37}$$

where $G_i = \max\big(1, \lceil |h_i|/(\Delta_J) \rceil\big)$ and $\Delta_J$ is a reference coupler scale (e.g., $\Delta_J = \text{median}_{i<j} |J_{ij}|$ or $\max_{i<j} |J_{ij}|$). Let $\sigma_i = \text{sign}(h_i)$. We implement the following bounded-scale Ising model on hardware:

$$E_{\text{split}}(s, a) = -\sum_{i<j} J_{ij} \, s_i s_j - \sum_{g=1}^{G} \Big( \sum_{i=1}^{n} \sigma_i \, w_{ig} \, s_i \Big) a_g - \sum_{g=1}^{G} B_g \, a_g, \tag{38}$$

with modest ancilla biases $B_g$ chosen by a simple rule such as

$$B_g = (1 + \epsilon) \sum_{i=1}^{n} w_{ig} \quad \text{with} \quad 0 < \epsilon \ll 1, \tag{39}$$

and then jointly rescaled with $\{J_{ij}\}$ to the device's representable range. By construction, each coupling $|\sigma_i w_{ig}|$ is of the same order as $\Delta_J$, so after the single global rescaling/quantization step the $J_{ij}$ entries retain substantially more effective resolution. In practice this "field-to-coupler splitting" acts as a precision-aware reparameterization: it reduces dynamic-range imbalance between $h$ and $J$ without altering the data–data couplers $\{J_{ij}\}$.

## E  IMPLEMENTATION DETAILS

We benchmark on five image–classification datasets, deliberately using relatively small batch sizes $b$ to emphasize the small–batch regime where diversity matters. Table 2 summarizes the datasets and our default AL settings. We largely follow prior configurations, but intentionally adopt relatively small batch sizes $b$ to highlight the importance of diversity in the small-batch regime. Unless otherwise noted, the backbone is ResNet-18 (He et al., 2016). Our DAL framework is built on the evaluation setup of (Zhan et al., 2022); please refer to their paper for additional architectural and training details.

$i$ is the size of initial labeled pool, $u$ is the size of unlabeled data pool, $t$ is the size of testing set, $b$ is the per-round batch size, $Q$ is the total query budget, $C$ is the number of categories, and $e$ is the number of epochs used to train the basic classifier in each AL round. Each algorithm is run three times per dataset to report the mean and variance. $M_{\max}$ is the maximum block size and $\tau$ is the candidate ratio (fraction of unlabeled pool sampled as candidates). For fairness, we set the number of training epochs for *Uherding* to match the other methods, while keeping its original learning rate $0.025$; empirically, reducing it to $0.001$ (the rate used by other methods) led to much slower accuracy improvements. Because the public *Uherding* codebase does not provide configurations for *SVHN*, *MNIST*, and *FashionMNIST*, we did not reproduce *Uherding* on these three datasets. The head-to-head comparison between the quantum solver and the greedy algorithm was conducted with batch size $b = 100$ and candidate ratio $\tau = 0.06$, reflecting the nontrivial overhead of invoking current quantum hardware and chosen to conserve computational resources.

When blocks are used, we fix the maximum block size $M_{\max}$, adopt proportional quotas ($\alpha_b = w_b$), and use the same candidate ratio $\tau$ to form UCPs. The kernel function we adopt is the Gaussian kernel function. Kernel bandwidths are set per block via the median heuristic; the solver for the MMD subproblem is identical across runs. Each method is run three times per dataset with different seeds.

| Dataset | $i$ | $u$ | $t$ | $b$ | $Q$ | $C$ | $e$ | $M_{\max}$ | $\tau$ |
|---|---|---|---|---|---|---|---|---|---|
| *MNIST* | 100 | 59,500 | 10,000 | 50 | 1,000 | 10 | 20 | 5,000 | 0.1 |
| *FashionMNIST* | 500 | 59,500 | 10,000 | 50 | 5,000 | 10 | 20 | 5,000 | 0.1 |
| *SVHN* | 500 | 72,757 | 26,032 | 50 | 5,000 | 10 | 20 | 5,000 | 0.1 |
| *CIFAR10* | 1,000 | 49,000 | 10,000 | 100 | 6,000 | 10 | 30 | 5,000 | 0.1 |
| *CIFAR100* | 1,000 | 49,000 | 10,000 | 100 | 10,000 | 100 | 40 | 5,000 | 0.1 |

Table 2: Datasets used in comparative experiments.

Figure 4 shows the measured expectation of the problem Hamiltonian during a single annealing run. The characteristic steep initial descent reflects the transition from the initial driver Hamiltonian to the problem Hamiltonian, during which the system rapidly explores the energy landscape and converges toward low-energy regions. The long, flat plateau that follows indicates the system has settled into a metastable state within a low-energy basin, which corresponds to the solution subset returned by the annealer. In our encoding, a lower Hamiltonian expectation corresponds directly to a smaller MMD objective. Therefore, the observed trajectory confirms that the quantum annealer successfully progresses to a high-quality solution.

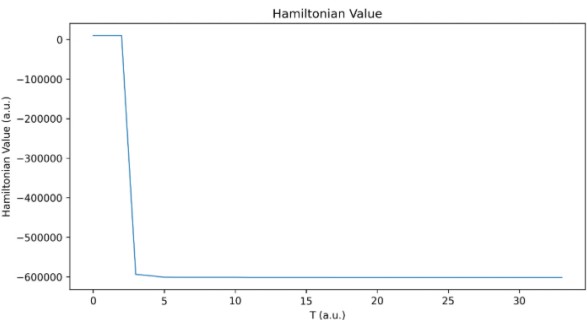

Figure 4: **Hamiltonian evolution.**

## F    ADDITIONAL PAIRWISE PLOTS (ABLATIONS)

In Figure 5 we present *pairwise* comparisons that isolate the effect of our diversity module: for each selector (Entropy, BALD, LPL) we plot its uncertainty-only variant against its "–EBDAL" counterpart under identical backbones, training schedules, and budgets. The only change within each pair is the addition of blockwise candidate filtering and MMD-based selection.

Across datasets, the EBDAL curves consistently *lift and smooth* the accuracy–budget trajectories relative to their baselines, with the largest gains on harder settings such as CIFAR-100. Improvements are already visible at small budgets—where uncertainty-only methods tend to sample clustered, redundant points—and remain evident as labeling proceeds on CIFAR-10, SVHN, MNIST, and FashionMNIST.

Overall, these plots confirm that coupling uncertainty with blockwise diversity reduces small-batch redundancy and improves label efficiency in a method-agnostic way.

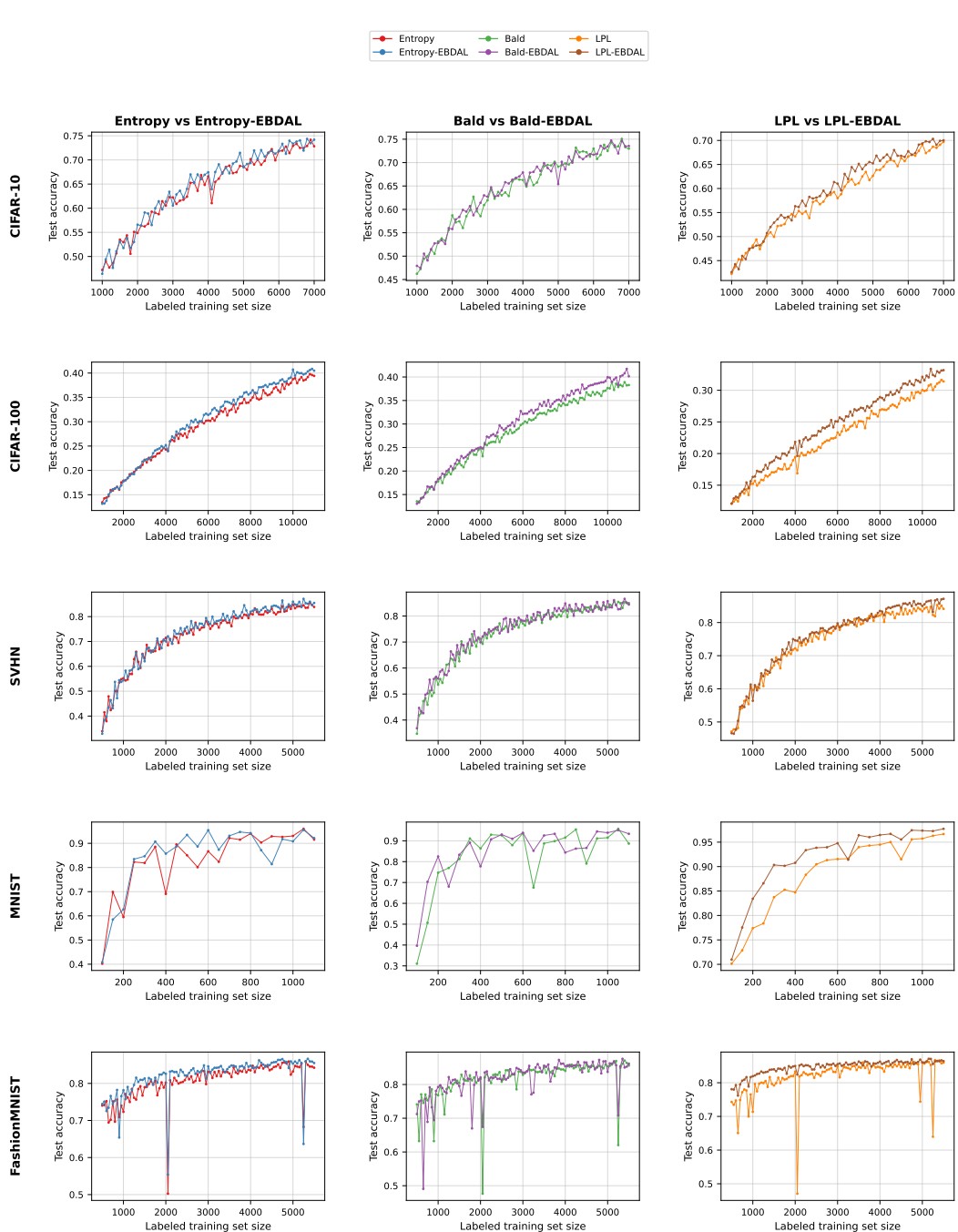

Figure 5: **Pairwise comparisons of uncertainty-only vs. EBDAL variants.** Columns: (left) Entropy vs. Entropy-EBDAL; (middle) BALD vs. BALD-EBDAL; (right) LPL vs. LPL-EBDAL. Rows: CIFAR-10, CIFAR-100, SVHN, MNIST, FashionMNIST. EBDAL smooths and elevates the curves at the same budget.