# OpenReview forum: "Efficient Blockwise Diverse Active Learning"
_ICLR.cc/2026/Conference — ICLR 2026 Conference Withdrawn Submission_

### Official Review · Reviewer_hxXJ · 2025-10-28

**Soundness:** 2
**Presentation:** 3
**Contribution:** 3
**Rating:** 2
**Confidence:** 4

**Summary:**

The paper introduces Efficient Blockwise Diverse Active Learning, or EBDAL, a framework that combines uncertainty-based sample selection with diversity to improve the efficiency of active learning. Instead of relying solely on uncertainty, EBDAL ensures representativeness by minimizing the Maximum Mean Discrepancy between the selected subset and the full dataset. The dataset is divided into clusters or blocks, and the method performs diversity-based selection within each block, which allows it to scale to large datasets while maintaining diversity. The MMD objective is formulated as a Quadratic Unconstrained Binary Optimization problem, enabling efficient greedy optimization and compatibility with quantum solvers. Theoretical analysis connects MMD minimization with data coverage, showing that the blockwise objective provides a close approximation to the global one.

**Strengths:**

The paper presents a clear and well-motivated integration of uncertainty and diversity, addressing a common problem in active learning where purely uncertainty-driven sampling leads to redundancy.

The theoretical development is rigorous and builds a strong link between MMD minimization and data coverage, providing an interpretable reason for why the method improves performance.

The inclusion of ablation studies makes the contribution of each component transparent.

**Weaknesses:**

The evaluation of the paper is not fully aligned with best practices in active learning research. As highlighted by [1] common pitfalls in active learning evaluation include:
* Lack of evaluated data distribution settings: the paper does not evaluate on imbalanced datasets.
* Lack of evaluated query sizes: the paper uses a fixed query size per dataset (50 for MNIST, FMNIST and SVHN, 100 for CIFAR10 and CIFAR100)
* Neglection of the classifier configuration: The paper does not have development datasets on which the hyperparameters were tuned, and roll out datasets where the performance is tested
* Neglection of alternative training paradigms: pre trained backbones were not tested.

Lack of baselines: the paper is lacking baselines like random sampling (no AL), margin selection [2], CoreSet[3], TypiClust [4], TCM [5].

The use of quantum solvers is only briefly demonstrated and lacks a detailed performance comparison, making it feel more like a side experiment than a core contribution.
Although the paper discusses computational complexity theoretically, empirical timing results are missing, leaving the real-world efficiency unclear.

The reliance on uncertainty filtering assumes well-calibrated uncertainty estimates, but the paper does not discuss how the method behaves under poor calibration. The K-means clustering used to create blocks is heuristic and may lead to uneven partitions, which can affect the approximation guarantees; no sensitivity analysis is provided to study this.

From a novelty standpoint, the theoretical contributions are adaptations of existing ideas from coreset.

Table 1 could include error bars or standard deviations to reflect variability.

No codebase provided, making it hard to reproduce the results.

[1] Lüth, Bungert, Klein, and Jaeger (2023) propose a systematic evaluation framework for active learning and analyze common pitfalls in AL literature.

[2] Bahri, Dara; Jiang, Heinrich; Schuster, Tal; Rostamizadeh, Afshin (2022). Is margin all you need? An extensive empirical study of active learning on tabular data. arXiv preprint arXiv:2210.03822.

[3] Sener, Ozan and Savarese, Silvio (2017). Active Learning for Convolutional Neural Networks: A Core-Set Approach. arXiv preprint arXiv:1708.00489.

[4] Hacohen, Guy; Dekel, Avihu; Weinshall, Daphna (2022). Active Learning on a Budget: Opposite Strategies Suit High and Low Budgets. In Proceedings of the 39th International Conference on Machine Learning (PMLR).

[5] Doucet, Paul; Estermann, Benjamin; Aczél, Till; Wattenhofer, Roger (2024). Bridging Diversity and Uncertainty in Active Learning with Self-Supervised Pre-Training (TCM). arXiv preprint arXiv:2403.03728.

**Questions:**

Does it outperform old and suggested baselines on new datasets, other query sizes and with pre-trained models?

How robust is EBDAL to miscalibrated uncertainty estimates, especially when uncertainty filtering is the first step of selection?

Can the authors provide empirical runtime comparisons between EBDAL and baselines to substantiate the claimed computational efficiency?

---

### Official Review · Reviewer_HHYM · 2025-10-29

**Soundness:** 3
**Presentation:** 3
**Contribution:** 2
**Rating:** 2
**Confidence:** 4

**Summary:**

The paper proposes a blockwise active-learning scheme that (i) filters a high-uncertainty candidate set per block, then (ii) selects within each block to minimize MMD between the selected subset and the pool. The MMD objective is rewritten as a QUBO; the authors argue submodularity of a corresponding quadratic set function permits a greedy method, and they also claim an "end-to-end" selection on real quantum hardware. Experiments on five small/medium image datasets show consistent AUC gains over uncertainty-only baselines, plus ablations.

**Strengths:**

1. The paper provides a formal approximation bound (Theorem 2) to justify the blockwise optimization of MMD.
2. The blockwise design is practical, offering a computationally efficient alternative to optimizing over the entire candidate set at once. The paper also provides a theoretical analysis of the error bound for this blockwise MMD approximation.
3. The empirical results across several datasets consistently support the effectiveness of the proposed method compared to the chosen baselines.

**Weaknesses:**

1. The core idea of combining uncertainty sampling with MMD-based diversity is well-established in the active learning literature [1-3]. Formulating the MMD objective as a quadratic program is also a known technique. The primary contribution appears to be the blockwise optimization, which feels more like an engineering improvement for efficiency rather than a fundamental algorithmic advance.

2. While the blockwise approach improves efficiency, it is a relatively standard heuristic for managing the high computational complexity of the full optimization. Furthermore, the motivation for selecting an equal number of points from each block is not well-justified. It's unclear why this constraint is beneficial or how it interacts with the underlying data distribution. It is more like to meet the condition in the theoretical analyses.

3. The rationale for objective (4), which appears to select a set with minimal training loss, is unconvincing. Minimizing training loss on a candidate set does not necessarily correlate with improved generalization error for the model. This objective seems counter-intuitive, as it might favor "easy" samples the model already understands, rather than informative ones that would improve its decision boundary.

4. The efficiency of the proposed method has not been well justified. The algorithm requires re-clustering the entire unlabeled pool $\mathcal{D}_U$ at every AL round (Algorithm 1, step 5). This K-means step has a high computational cost. Besides, a comprehensive wall-clock time benchmark for a full acquisition step is missing (Algorithm 1, steps 4-13), including uncertainty calculation, the $O(N)$ clustering step, kernel computation, and the final selection.

[1] Chattopadhyay, Rita, et al. "Batch mode active sampling based on marginal probability distribution matching." Proceedings of the 18th ACM SIGKDD international conference on Knowledge discovery and data mining. 2012.

[2] Wang, Zheng, and Jieping Ye. "Querying discriminative and representative samples for batch mode active learning." Proceedings of the 19th ACM SIGKDD international conference on Knowledge discovery and data mining. 2013.

[3] Wang, Zengmao, et al. "Incorporating distribution matching into uncertainty for multiple kernel active learning." IEEE Transactions on Knowledge and Data Engineering 33.1 (2019): 128-142.

**Questions:**

Why was the Uherding (Bae et al., 2025) baseline, which is cited as recent work, omitted from the experiments on MNIST and FashionMNIST in Table 1?

---

### Official Review · Reviewer_61SX · 2025-10-30

**Soundness:** 2
**Presentation:** 3
**Contribution:** 2
**Rating:** 2
**Confidence:** 4

**Summary:**

This paper proposes **Efficient Blockwise Diverse Active Learning (EBDAL)**, which combines uncertainty filtering with a blockwise MMD-based selection strategy. The core idea is to partition the unlabeled pool into several blocks, solve a QUBO-formulated MMD diversity objective in each block, and aggregate the selected samples to approximate global coverage with reduced complexity (`O(sum n_b^2)`).
The paper presents theoretical analyses connecting coverage probability `ρ(r)` with a cross-term `δ_MMD`, a blockwise approximation theorem for global MMD, and empirical results showing faster runtime and comparable accuracy to full MMD selection.

**Strengths:**

- Proposes a creative integration of uncertainty filtering and diversity sampling.
- The blockwise formulation is computationally appealing and intuitive.
- The presentation and visual layout are professional and easy to read.

**Weaknesses:**

- Multiple inconsistencies across Lemma 1, Theorem 1, and Theorem 2. Key terms are undefined, and some inequalities are mathematically invalid or too loose.
- The experimental setup is weak and does not convincingly support the efficiency or effectiveness claims.
  - Datasets are small (CIFAR-10, SVHN), lacking large-scale or non-visual tasks.
  - Performance improvements are marginal (1–2%) with overlapping standard deviations.
  - Baselines are outdated.
  - No ablation study to isolate the effects of blockwise partitioning, uncertainty filtering, or QUBO optimization.
  - Efficiency claims lack reproducibility (no batch-size, cache, or memory metrics; “0.2 ms per quantum call” is unsupported).
-  Theorems assume fixed kernel space, while implementation uses embedding space.
- The paper presents heuristic ideas as formal theorems, which overstates the level of rigor.

**Questions:**

1. Please clarify the relationship between `p` and `1−ε` in Lemma 1. Are they independent parameters or linked via coverage probability?
2. Can the authors define `δ_MMD` explicitly and show conditions ensuring `ρ(r) ≤ 1`?
3. Why is the factor “2” required in Theorem 2? Would `MMD^2(S,X) ≤ Σ_b w_b MMD^2(S_b,X_b)` not suffice?
4. Under what settings (λ range, normalization) is the QUBO objective truly submodular?
5. Are the theoretical results intended to apply to the embedding space used in experiments? If so, how does varying `σ_b` affect the derived bounds?
6. Could the authors provide reproducible runtime measurements, hardware configurations, and statistical tests to validate the efficiency claims?

---

### Official Review · Reviewer_yb4g · 2025-10-31

**Soundness:** 2
**Presentation:** 3
**Contribution:** 2
**Rating:** 4
**Confidence:** 2

**Summary:**

This paper introduces EBDAL (Efficient Blockwise Diverse Active Learning), a framework that explicitly integrates both uncertainty scores and diversity scores when selecting samples for active learning.   EBDAL’s contribution using splitting the dataset into blocks to speed up processing and contribute toward diversity and ensuring diversity within each block.  Diversity is quantified by using Maximum Mean Discrepancy (MMD) within each block.  The required computations are formalized as a Quadratic Unconstrained Binary Optimization (QUBO) problem, whose objective function is submodular.  A bounded approximation guarantee showing that blockwise MMD closely tracks the global MMD, providing a theoretical foundation for the approach.  Experimental results are given on standard image datasets, including implementing the QUBO algorithm on a quantum computer.

**Strengths:**

- a bounded approximation guarantee showing that blockwise MMD closely tracks the global MMD,  which provides a good theortical foundation for the algorithm

- investigating a quantum QUBO implementation of EBDAL

- The EBDAL algorithm was clearly described

**Weaknesses:**

- the related work should have included a better description of other active learning algorithms that integrate uncertainty scores and diversity scores and how EBDAL differs (perhaps VAAL, SIMILAR, BMDR, etc.)

- the connection with quantum computing implementations of active learning in general and QUBO/Ising in particular did not seem to me to be well integrated with the rest of the paper, but almost an after thought

- in several of the experiments the advantages of EBDAL seem relatively minor compared to the baseline (* vs *-EBDAL) and given the complexity of implementing EBDAL, it wasn't obvious to me that the extra effort was justified

**Questions:**

- I think the paper would be stronger if the quantum implementation of the QUBO algorithm was described with a bit more context and more detail was provided.  As presented it seemed almost an afterthought, not one of the main contributions as explained (line 82-85)

- one of the main contributions is described (lines 73-74) "substantially reduces small-batch redundancy," but I didn't see in the paper how that was quantified

---

### Note · Authors · 2025-11-17

I have read and agree with the venue's withdrawal policy on behalf of myself and my co-authors.